# Childbearing with Hypermobile Ehlers–Danlos Syndrome and Hypermobility Spectrum Disorders: A Large International Survey of Outcomes and Complications

**DOI:** 10.3390/ijerph20206957

**Published:** 2023-10-21

**Authors:** Gemma Pearce, Lauren Bell, Sally Pezaro, Emma Reinhold

**Affiliations:** 1Research Centre for Healthcare and Communities, Coventry University, Coventry CV1 5FB, UK; lauren.bell@coventry.gov.uk (L.B.); sally.pezaro@coventry.ac.uk (S.P.); 2Coventry City Council, Coventry CV1 2GN, UK; 3Patient and Public Involvement and Medical Advisor, Coventry University, Coventry CV1 5FB, UK; e.reinhold@doctors.org.uk

**Keywords:** hypermobility, Ehlers–Danlos, pregnancy, birth, maternity, incidence, women’s health and wellbeing, labour, Mast Cell Activation syndrome (MCAS), premature birth

## Abstract

One in 20 births could be affected by hypermobile Ehlers–Danlos syndrome or Hypermobility Spectrum Disorders (hEDS/HSD); however, these are under-diagnosed and lacking research. This study aimed to examine outcomes and complications in people childbearing with hEDS/HSD. A large online international survey was completed by women with experience in childbearing and a diagnosis of hEDS/HSD (*N* = 947, total pregnancies = 1338). Data were collected on demographics, pregnancy and birth outcomes and complications. Participants reported pregnancies in the UK (*N* = 771), USA (*N* = 364), Australia (*N* = 106), Canada (*N* = 60), New Zealand (*N* = 23) and Ireland (*N* = 14). Incidences were higher in people with hEDS/HSD than typically found in the general population for pre-eclampsia, eclampsia, pre-term rupture of membranes, pre-term birth, antepartum haemorrhage, postpartum haemorrhage, hyperemesis gravidarum, shoulder dystocia, caesarean wound infection, postpartum psychosis, post-traumatic stress disorder, precipitate labour and being born before arrival at place of birth. This potential for increased risk related to maternal and neonatal outcomes and complications highlights the importance of diagnosis and appropriate care considerations for childbearing people with hEDS/HSD. Recommendations include updating healthcare guidance to include awareness of these possible complications and outcomes and including hEDS/HSD in initial screening questionnaires of perinatal care to ensure appropriate consultation and monitoring can take place from the start.

## 1. Introduction

When thinking about starting a family, people with long-term medical conditions have an additional layer of considerations around health and risk to themselves and their baby. The Ehlers–Danlos syndromes (EDS) are a heterogenous group of genetic disorders affecting the structure, processing and functioning of collagen [1,2] with features that may have an impact on childbearing [3]. Most sub-types of EDS such as classical and vascular are rare; however, the hypermobile subtype (hEDS) [4] is understood to be more common [5] (see Box 1 for diagnostic criteria and nomenclature history).

Box 1Diagnostic criteria and nomenclature history of hEDS/HSD.Although hEDS is inherited in an autosomal dominant fashion, work to identify the genetic cause is ongoing [6]. Diagnosis is currently based on the fulfilment of clinical criteria [1] published in 2017 and based on expert consensus but without prior validation. They consist of generalized joint hypermobility (criterion 1), systemic manifestations of a generalised connective tissue disorder, family history of hEDS, and/or musculoskeletal complications (criterion 2) and exclusion of alternative diagnoses (criterion 3). Individuals who do not meet the full hEDS criteria but who exhibit similar signs and symptoms may be diagnosed with a Hypermobility Spectrum Disorder [1,7]. Although a “spectrum of symptoms” has been described, it is now understood that people with either hEDS or HSD may experience the same potential range of symptoms [8], complications and disease severity [9], and recommended treatments and prognosis are the same; thus, the distinction may not be clinically meaningful [10,11]. Prior to the 2017 criteria [1], previous (and overlapping) [12] diagnostic classifications were EDS Type III [13], EDS—Hypermobility Type [14] or Joint Hypermobility Syndrome (JHS) [15]. Since 2017, individuals should be deemed to have hEDS or HSD without the need for reassessment, with the exception of clinical trial entry. This study therefore refers to hEDS/HSD, inclusive of previous names for these diagnoses, and in awareness that the nomenclature is likely to change in the future.

The first study of the diagnosed prevalence of hEDS/HSD found that 1 in 500 people in Wales had a diagnosis of either EDS or Joint Hypermobility Syndrome (JHS) on their medical records [5]. Other work has shown figures of 1 in 225 [16] and 1 in 830 [17] for recorded diagnosis. We know that many people face long diagnostic delays [18,19], with many undiagnosed [20]. A total of 3.4% of the general population have chronic pain and hypermobility, which has been used as a proxy for hEDS/HSD [21,22], and led to the estimate that 1 in 20 births could be impacted by hEDS/HSD [23]. In healthcare, hEDS/HSD can be inadequately managed because of the underdiagnosis and insufficient awareness of these conditions [24,25,26,27], as well as their heterogenous clinical presentation [28]. Understanding the implications for childbearing among people with hEDS/HSD is therefore essential to achieve a positive and safe birth experience [29,30]

Guidance recommends that an individual assessment is carried out for planning birth with a connective tissue disorder [31]. It is therefore important that perinatal staff and individuals with hEDS/HSD are aware of the potential risks of complications associated with childbearing. However, there are limitations to the existing evidence base, including the fact that research has not always differentiated between hEDS and the other rare subtypes of EDS. To illustrate, a population-based study found higher rates of premature birth, birth by caesarean section and several other complications, along with longer hospital stays for post-partum care [32]. However, these findings incorporated all types of EDS, including the vascular subtype known to have serious risks during pregnancy [33].

Existing research is inconsistent about the rates of obstetric complications among people with hEDS/EDS. Increased rates of miscarriages (including specifically recurrent miscarriage) [34], preterm birth [35] and pelvic instability [36] have been reported, whilst a large records-based study found pregnancy complication rates similar to population incidences [37]. A small but in-depth study of women with JHS found “good” overall outcomes, but potential issues with abnormal scar formation, haemorrhage, pelvic prolapses, deep venous thrombosis and coccyx dislocation [38]. Precipitate labour (the expulsion of the foetus within three hours of the commencement of contractions) [39] has also been suggested as possibly more prevalent among people with hEDS/HSD [23,38] and is associated with significant complications [40]. Qualitative research has reported experiences of the worsening of hEDS/HSD symptoms during pregnancy, as well as ineffective local anaesthesia and long latent phases of labour followed by rapid births [23]. Contemporary evidence with larger sample sizes following the 2017 re-classification [1] is now required to enhance the quality and safety of evidence-based practice. The aim of this study was to examine pregnancy and birth outcomes in people with hEDS/HSD through a large-scale international survey, to provide a new understanding of the full range of complications and thereby inform perinatal care and increase obstetric safety.

## 2. Materials and Methods

An online international survey was carried out predominantly collecting quantitative data with additional open-ended qualitative questions to allow participants to clarify responses. Five third-party organisations (Royal College of Obstetricians and Gynaecologists, The Better Birth Network, Ehlers–Danlos Society, Hypermobility Syndromes Association and Ehlers–Danlos Support UK) were involved in the development of this project, as partners in the funding application and supported recruitment. This was the first part of a larger project that also explored childbearing women and healthcare professionals’ experiences of perinatal care, followed by the co-creation of useful tools for implementation in practice [41].

Participants recruited were women over the age of 18 years who had given birth after 24 weeks of pregnancy at least once from 2007 onwards (though participants meeting this criterion could also report on all pregnancies after 2007). The timeframe chosen was to ensure that birthing experiences were after the publication of the World Health Organisation recommended interventions for improving maternal and newborn health [29]. Participants were required to declare a medical diagnosis of hEDS, HSD or an older equivalent diagnosis of EDS Type III, EDS—Hypermobility Type or Joint Hypermobility Syndrome. Inclusion was limited to developed, majority native English-speaking countries; namely UK, Ireland, USA, Canada, Australia and New Zealand [42]. Participants were recruited through public, professional, voluntary sector and social media platforms between June and September 2019.

The survey questions were developed following a review of the available literature in relation to childbearing with hEDS/HSD and modified following piloting and ethical review (e.g., questions were prioritised to reduce survey length). Patient and public involvement (PPI) was conducted via an online poll and led to the inclusion of certain complications—for example, premature birth, abnormal length of labour/birth, poor anaesthetic coverage and haemorrhage (see Pearce et al., 2023 for further detail of PPI [41]). To facilitate the understanding of terms (e.g., augmentation and induction of labour), definitions were provided to participants and are reported here alongside the results. The survey was hosted on Qualtrics software starting with participant information and consent before the completion of the survey. Complex survey logic was implemented so participants were not asked to complete inappropriate or unethical questions (e.g., after reporting a miscarriage). Participants could withdraw by closing the browser at any time, without reason or consequences, and they could pause participation by using the ‘Save Later’ option and return to complete the survey within one week. Participants inputted a code of their choice so they could withdraw by emailing the researcher up until the point of data analysis. Participants reported their demographics (age, ethnicity, level of education) and, for each pregnancy, the outcome, gestational week, lengths of birth stages, place and mode of birth, and then selected which birth complications they had experienced from a list (see Table 1 in the results section). Participants repeated the questions for each pregnancy since 2007 and could provide any further comments about their childbearing experience via an open text box. A debrief page was provided, signposting participants to sources of information and support.

Data were analysed with SPSS 28 software using descriptive statistics to two decimal places. Data were reviewed to ensure all responses met eligibility criteria, and descriptive statistics were calculated for participant demographics. Percentage incidences were calculated for outcomes and complications (e.g., pre-eclampsia, pre-term birth, shoulder dystocia, haemorrhages, ineffective pain relief). Where responses were binary, i.e., experienced or not experienced, 95% confidence intervals for proportions were calculated. Survey data are presented alongside relevant published general population incidences, collated where available following literature review, to aid interpretation in the context of the study design.

All qualitative responses were read for familiarity and analysed using conventional content analysis to compliment the quantitative focus and capture additional complications not asked about quantitatively. This approach is appropriate when limited literature exists and to analyse alongside descriptive statistics [43,44]. To increase reliability, two members of the research team were collectively involved in coding and analysis, with comments coded independently and then discussed with the second researcher (GP, LB). Qualitative codes and associated quotes relating to complications of childbearing were grouped into themes. Themes were developed inductively from text, rather than from the complications directly asked about in the survey. Findings were discussed in light of perinatal care (SP) and medical expertise with lived experience (ER). Code frequencies are not reported with findings, as qualitative data were collected to supplement quantitative data and cannot infer the prevalence of a complication. Any identifiable information in open-ended responses was anonymised.

## 3. Results

The number of surveys completed was 955. Data for outcomes and complications are based on 947 responses, because eight respondents appeared to enter data on four separate pregnancies in one single round of the survey and so were excluded from analysis. A total of 1338 pregnancies were reported with outcomes as reported in Figure 1. The majority of participants (Mean age of respondents = 35.71 years, SD = 6.84) were white (93%) and educated with high school (*N* = 194, 20.49%), further education (*N* = 170, 17.95%), higher education (*N* = 470, 49.63%), professional qualifications (*N* = 33, 3.48%) or equivalent other (*N* = 66, 6.97%). Just under half (47.94%) of the participants had received a diagnosis based on the more recent 2017 criteria (hEDS = 375, HSD = 79), while the remainder had diagnoses from older criteria for these conditions (EDS type III = 179, EDS-HT = 131, JHS = 183). At least 65% of participants conceived their first pregnancy before receiving a diagnosis. Most pregnancies took place in the UK (*N* = 771, 57.62%), with the rest in the USA (*N* = 364, 27.20%), Australia (*N* = 106, 7.92%), Canada (*N* = 60, 4.48%), New Zealand (*N* = 23, 1.72%) and Ireland (*N* = 14, 1.05%). The majority (77.20%) were planned pregnancies, with 787 out of 1033 conceived in less than 12 months.

### Pregnancy and Birth Outcomes and Complications

Most were singleton pregnancies (98.01%) with the remainder being twins (Figure 1). The majority of participants reported their births to be in a consultant-led obstetric unit (*N* = 946, 77.67%). The remaining births were reported as being in maternity units led by midwives (*N* = 207, 17.00%), home births (*N* = 42, 3.45%) or births before arriving at the intended place of birth (*N* = 23, 1.89%).

In relation to babies born after 24 weeks (*N* = 1230), 190 (15.45%) were pre-term (occurring before 37 weeks) including 7 stillbirths. Of the 183 pre-term live births, 149 were singleton pregnancies (80 vaginal births; 69 caesarean births) and 34 were twins (7 vaginal births; 27 caesarean births). There were 44.80% (*N* = 551; vaginal = 358; caesarean = 193) births at term before the due date (between ≥37 weeks and <40 weeks) and 39.76% (*N* = 489; vaginal = 402; caesarean = 87) births after the due date (≥40 weeks) (Figure 2). These data are shown in Figure 2 alongside total births by gestational age from the England and Wales Office for National Statistics [45]. This figure highlights that pre-term births and births from 42 weeks were higher for the participants with hEDS/HSD and lower between 39–41 weeks, compared with these general population statistics [45].

Out of the 1041 pregnancies with live or stillborn vaginal births and emergency caesareans (i.e., excluding planned caesarean births), induction of labour, described to participants as inserting a tablet (pessary) or gel into the vagina, was reported in 153 (14.70%). Augmentation of labour, defined as drugs to speed up the birth of the baby after contractions had started, was reported by 135 (12.97%). A quick or abnormally rapid labour was reported in 357 (34.29%) births, defined to participants as the birth of a baby within less than 3 h of the start of regular contractions.

Table 1 presents the data on the outcomes and complications of pregnancy and birth as reported by the participants with population incidences from the literature, where available. This was based on the number of pregnancies that ended as either live or still born births in the third trimester and therefore excluded miscarriage, ectopic pregnancies and terminations (*N* = 1209)—for example, where the births of twins vaginally would be counted as one pregnancy. Outcomes and complications that have been highlighted in bold in the table indicate where confidence interval findings from this study sit outside of the general population incidences.

Out of the 1209 pregnancies, 43.67% (*N* = 528) reported their latent phase of labour length. The data on latent labour were not normally distributed, as indicated on a histogram, with reports of occurrences that lasted weeks, and 148 (12.24%) reporting latent labours longer than 24 h. For those that did not report a latent phase, some added statements to explain that they did not know how long it was, were not aware of their latent phase, could not remember the change from latent to active labour or it was not applicable to them because they had a caesarean section. Some participants added qualitative text further describing long latent phases and quick births that were not always in their planned place, for example “*2 days [latent phase]. 3 min from 4cm to birth though*”. and “*Onset of labour was so quick the baby was born 7 min after I was discovered to be in active labour. Almost was born in the elevator between the inpatient and birthing floor*”.

**Table 1 ijerph-20-06957-t001:** The outcomes and complications of pregnancy and birth.

Complications/Outcomes	Pregnancies in Third Trimester (*N* = 1209) ^1^	Percentage of Pregnancies (95% Confidence Intervals)	Published Population Incidence
**Hyperemesis Gravidarum**	308	**25.48% (23.04–28.03)**	**3%** [46]
**Antepartum Haemorrhage**	104	**8.60% (7.08–10.33)**	**3–5%** [47]
**Pre-term Rupture of Membranes**	85	**7.03% (5.65–8.62)**	**<3%** [48]
**Pre-eclampsia**	115	**9.51% (7.92–11.31)**	**2.4%** [49]
**Eclampsia**	17	**1.41% (0.82–2.24)**	**0.027%** [50]
**Preterm birth < 37 weeks (singleton)**	155/1185	**13.08% (11.21–15.13)**	**6.2%** [51]
**Preterm birth < 37 weeks (twin)**	35/46	**76.09% (61.23–87.41)**	**50%** [51]
**Preterm birth < 37 weeks (all)**	190/1230	**15.45% (13.47–17.59)**	**7.8%** [51]
Stillbirth ≥ 24 weeks (singleton)	6/1185	0.51% (0.19–1.10)	0.4% [51]
Stillbirth ≥ 24 weeks (twin)	1/46	2.1% (0.05–11.5)	2.3% [51]
Stillbirth ≥ 24 weeks (all)	7/1230	0.57% (0.23–1.17)	0.48% [51]
Abnormal foetal presentation	338	27.96% (25.44–30.58)	N/A
**Precipitate labour (<3 h)**	357/1041	**34.29% (31.41–37.27)**	**0.07–14%** [39,40]
**Born before arrival at intended place of birth**	23	**1.90% (1.21–2.84)**	**0.4%** [52]
Caesarean section	363	30.02% (27.45–32.70)	**27.6%** [53]
**Shoulder dystocia**	52/847	**6.14% (4.62–7.97)**	**0.18%** [54]
1st degree perineal tear	170/847	20.07% (17.42–22.93)	N/A
2nd degree perineal tear	303/847	35.77% (32.54–39.11)	N/A
3rd degree perineal tear	72/847	8.50% (6.71–10.59)	N/A
4th degree perineal tear	17/847	2.01% (1.17–3.19)	N/A
Intrapartum haemorrhage	173	14.31% (12.38–16.41)	N/A
**Postpartum haemorrhage**	249	**20.60% (18.35–22.99)**	**6.38%** [55]
Ineffective pain relief	517	42.76% (39.95–45.61)	N/A
Ineffective epidural	284	23.49% (21.13–25.98)	8–23% [56]
Ineffective local injection	192	15.88% (13.86–18.07)	N/A
**Caesarean wound infection**	87/363	**23.97% (19.67–28.70)**	**3–15%** [57]
Vaginal wound infection	61/847	7.20% (5.55–9.16)	N/A
Wound dehiscence	132	10.92% (9.22–12.81)	N/A
Slow healing	487	40.28% (37.50–43.11)	N/A
Pelvic organ prolapse	149	12.32% (10.52–14.31)	N/A
**Postpartum psychosis**	57	**4.71% (3.59–6.07)**	**0.089–0.26%** [58]
**Post Traumatic Stress Disorder**	227	**18.78% (16.61–21.09)**	**4%** [59]
Other	87	7.20% (5.80–8.80)	N/A
None	71	5.87% (4.61–7.35)	N/A

Note. The outcomes and complications that have been put in bold highlight where confidence interval findings from this study sit outside of the general population incidences. ^1^ Where the denominator differs from 1209, this is explicitly provided in the table. Where *N* = 847, this is a total of vaginal births only. Where *N* = 363, this is a total of caesareans only. Where *N* = 1230, this is a total of babies birthed rather than pregnancies (for example, where 2 vaginally birthed twins are counted as 2 rather than 1). This represents 1223 live births plus the 7 stillbirths, giving totals for singleton (*N* = 1185) and twin (*N* = 46) births in the table. The total of 1041 relates to pregnancies (not births) ending in vaginal births and emergency caesareans.

Where participants selected ‘other’, listed complications included gestational diabetes (*N* = 7, 0.58%), Haemolysis, Elevated Liver enzymes and Low Platelets (HELLP) syndrome (*N* = 6, 0.50%), obstetric cholestasis (*N* = 7, 0.6%), retained placenta (*N* = 6, 0.50%) and placenta previa (*N* = 4, 0.33%). There were five themed areas from the qualitative analysis: joints moving out of place and pain; issues from epidural; bruising, tearing and poor wound healing; blood pressure and syncope issues; and infections and unexplained allergic reactions.

(1)Joints moving out of place and pain: “hip dislocation in labour”, “sporadic pelvic displacement, wheelchair from 5 months until 2 weeks after”, “widespread post-partum joint pain and weakness”, “hip subluxation from stirrups (wasn’t aware of hypermobility)”, “severe looseness of shoulders and hips”, “pain, pain, pain”, “pelvic separation leading to bad post-partum pain”, “abdominal divarication, torn ligaments, stretched ligaments, pubis symphysis”; “Pubic Bone Separated at 20 weeks gestation”, “subluxed hips”, “costochondritis as baby moved from under ribs—misdiagnosed as indigestion at the time (EDS not diagnosed at that time)”, “pelvic displacement”, “SI *[sacroiliac]* joint issues, headaches” and “induced labour caused hip dislocations”.(2)Issues from epidural anaesthesia: “CSF *[cerebral spinal fluid]* leak”, “post-dural puncture headache from a ‘technically difficult’ epidural (x3) requiring a blood patch *[sic]*”, “CSF leak from epidural and have continued pain there since”, “extreme hypotension from epidural”; “hypotension with epidural. Prolonged effects of epidural”, and three people said “broken” or “dislocated coccyx”.(3)Bruising, tearing and poor wound healing: “very significant bruising”, “bruising on the baby from fast birth”, “site of vaginal tear never fully healed”, “previous Caesarean scar ruptured during this labour”; “polyps in birth canal from tearing”, “torn hip labrum from long labour” and “bilateral femoral nerve damage and bilateral hip labral tears”.(4)Blood pressure and syncope issues: “blood pressure dropped severely (PoTS) [Postural (orthostatic) Tachycardia Syndrome]”, “tachycardia”, “syncope several hours following birth”, “low blood pressure” and “high blood pressure after delivery”.(5)Infections and unexplained allergic reactions: “unexplained anaphylaxis 7 days and 8 days postpartum respectively [*sic*]”, “chorioamnionitis”, “systemic infection”, “baby born with group B strep”, “infection due to time between waters breaking and giving birth”, “bladder infection”, and four people said “uterine infection”.

## 4. Discussion

The aim of this study was to explore the outcomes and complications of pregnancy and birth for people with hEDS/HSD. A large number of participants completed the survey in a short space of time, highlighting the desire for people with hEDS/HSD to extend the knowledge base for these conditions via research. In summary, the results confirmed higher incidences of certain childbearing complications than are typically found in the general population and reports others for the first time. These include preterm birth, pre-eclampsia, eclampsia, Pre-term Rupture of Membranes, antepartum haemorrhage, postpartum haemorrhage, hyperemesis gravidarum, shoulder dystocia, caesarean wound infection, postpartum psychosis, PTSD, precipitate labour and being born before arrival at intended place of birth. Higher incidences are concluded whereby ranges for population incidences fall entirely outside of the confidence intervals found in this study. The meaning and implications of these findings are subsequently discussed.

This study corroborates previous findings with roughly twice the population incidence of preterm birth among people with hEDS/HSD [35]. In addition, this study found higher than typical rates of pre-term rupture of membranes, a complication associated with neonatal morbidity and mortality, alongside additional risks from chorioamnionitis and placental abruption [48]. It could be that this fragility of foetal membranes is one of the factors that contributes to the higher incidence of prematurity. Findings did not reveal any increased incidence of stillbirth, consistent with previous research on JHS [37]. Whilst previous research has found a higher rate of miscarriage among people with hEDS [34] than has been reported in the general population [60], this study found no increased incidence.

Antepartum haemorrhage, sometimes caused by placental abruption, was higher than would be expected based on population incidences [47]. Bleeding is not only a risk in itself, but placental abruption has been associated with pre-eclampsia [61], with both eclampsia and pre-eclampsia reported at much higher rates in this study than in the general population [49,50]. Identifying risk factors for pre-eclampsia (e.g., pre-existing hypertension, having pre-eclampsia in a previous pregnancy) [62] is embedded in perinatal care. This work suggests that people with hEDS/HSD should also be considered as an at-risk group. It is recommended that findings from this hEDS/HSD population are included in future updates of clinical guidelines [63] and that hEDS/HSD are embedded in initial pregnancy screening questionnaires. Researchers may further investigate the mechanisms that lead to pre-eclampsia. For example, magnesium is one of the important treatments for severe pre-eclampsia and eclamptic seizures, and magnesium depletion has been anecdotally linked to hEDS [64]. Other potential mechanisms could relate to mast cell degradation products causing issues with placental vasculature, blood flow or clotting, or the potential role of natural killer cells in pre-eclampsia, recurrent pregnancy loss and implantation failure [65,66].

A much higher than typical number of participants reported a precipitate labour [39,40], consistent with previous smaller hEDS/HSD studies [23,38]. This finding may help to explain the higher proportion of babies born before arrival at the intended place of birth [52] and warrants further longitudinal research to unearth the reasons and order of events. Labours of 3 h or less are strongly associated with higher risks of maternal complications including cervical and grade 3 perineal tears, post-partum haemorrhage, retained placenta and prolonged hospitalisation [40]. These risks are also important to include in perinatal care considerations, particularly given potential issues with wound healing and local anaesthetic for people with hEDS/HSD [4,67], although the incidence of ineffective local anaesthetic in this study was lower than previously found in dental care research [68]. Prolonged latent phases of labour may quickly progress into fast active labours and births for people with hEDS/HSD, with the subsequent risk that births are unattended by professionals. This outcome may contribute to a more negative birthing experience [69], as well as contribute to explaining the higher prevalence of pelvic organ prolapse (vaginal wall or womb) [70,71]. It should also be noted that participants in this study were only asked to report prolapses occurring in the first 6 weeks following birth, whereas prolapses are often not recognised until much later. To mitigate the potential risk of pelvic organ prolapse when episiotomy is clinically indicated, some authors have suggested that caesarean section may be preferred [72], though with its own risks. This study reports caesarean wound infection incidences as higher than the available population incidences. Slow healing was one of the most reported complications (by 40.28%); however, comparison to the general population is lacking and more research is required.

Shoulder dystocia was reported at a much higher incidence than the population average [54]. The known risk factors are maternal diabetes, body mass index over 25, age over 40, gestational age after 40 weeks, instrumental vaginal birth [54] and epidural, although its occurrence is still often unpredictable [73]. These findings indicate that hEDS/HSD may be another risk factor to consider and is therefore recommended to be considered in the next revisions of the guidance on shoulder dystocia [74]. The McRobert’s Manoeuvre is commonly recommended as first-line treatment [74,75]; however, this can result in maternal pubic symphysis diastasis [76] and other potential pelvic and hip injury. The Gaskin Manoeuvre in a hands-and-knees position has recently been found to be a safer first-line method reducing injury to the baby [77]. This could be considered a better first option for people with hEDS/HSD, warranting further research.

Postpartum haemorrhage was reported to be three times more common among surveyed participants than the general population incidence [55]. This has profound impact as a leading cause of maternal death often caused by failure of the uterus to contract back adequately after birth, genital tract trauma or retained placenta [55]. As hEDS/HSD has been linked to Mast Cell Activation syndrome (MCAS) [78,79,80], this finding might align with the theory that people with MCAS are more at risk of post-partum haemorrhage because of endogenous heparin release from mast cells or enhanced fibrinolysis driven by activated mast cells [81,82]. There is an overlap of reported birth complications between hEDS/HSD and MCAS, and therefore treatments for MCAS (e.g., identifying and removing environmental triggers, and the use of mast cell stabilisers and antihistamines) may be indicated alongside mainstream perinatal care [82]. Research and recognition of MCAS in pregnancy is currently lacking [82], yet it is believed to impact 35.9 million pregnancies worldwide annually [83]. This study supports the clear need for further examination of this complex interplay.

Although nausea and vomiting are estimated to occur in 50–90% of pregnancies, a much higher incidence of hyperemesis gravidarum was reported in this hEDS/HSD population (25.48%) compared to the general population (3%) [46]. hEDS/HSD has been referred to as a “trifecta” with MCAS and manifestations of autonomic dysfunction such as Postural (orthostatic) Tachycardia Syndrome (PoTS) because of high rates of co-occurrence between these conditions [84]. Given that there is also an increased risk of hyperemesis gravidarum in women with PoTS, potential mechanisms might involve a dysfunction of the autonomic nervous system [85]. Hyperemesis gravidarum is a prominent cause of emergency department visits and hospitalisation in pregnancy [86], and so this finding highlights the importance of diagnosing hEDS/HSD, supporting knowledge among perinatal staff and investigating the possible mechanisms implicated (e.g., Mast Cells), thereby potentially developing novel treatments.

Several studies have reported that people with hEDS/HSD do not feel listened to, contributing to traumatic experiences in maternity [23] and healthcare services [26,27,87]. Post-traumatic stress disorder (PTSD) was reported by almost 1 in 5 people in this study, consistent with the prevalence of PTSD previously found among high-risk groups (e.g., women who experienced an emergency caesarean section, or a difficult, traumatic or pre-term birth) [59]. Post-partum psychosis was also notably higher in this hEDS/HSD population at nearly 5%, compared with an estimated global prevalence of 1 in 1000 to 2000 births [58]. Further research that investigates this finding could facilitate advances in knowledge about the causes and treatments of this serious and distressing condition, with consideration needed as to whether individuals with hEDS/HSD ought to be included in existing monitoring systems.

### Strengths, Limitations and Implications

These data cover a large number of pregnancies (*N* = 1338) in over six countries and three continents, inclusive of up-to-date diagnostic criteria for hEDS/HSD. Population health surveys are useful for estimating condition prevalence [88], and this research specifically provides novel insights into perinatal complications and outcomes that may need additional considerations by contextualising them with reported incidences in the general population. Comparison statistics were chosen with preference given to recently published large empirical studies, reviews or guidelines from a similar context (e.g., high income country) and international variation in outcomes such as gestational age have been reported [89]. However, comparator data were not available for all complications or nations, and these statistics cannot be interpreted as a control group. This examination does provide a clear need for future investigation—for example, to collect and analyse large scale coded data, to document longitudinal childbearing journeys with hEDS/HSD and to consider potential mechanisms for these associations.

The eligibility criteria and self-selection of volunteer participants should be considered in the context of the findings being based on retrospective self-report data. To reduce recall bias and sit within current guidance [29], the cut off year for pregnancies was 2007; however, this means that the maternity history provided may only be partial for some individuals. Although participants reporting a birth after 24 weeks could report miscarriages, individuals who have only experienced miscarriages were not eligible to take part. Regarding fertility, a planned conception of pregnancy within 12 months (76%) was lower than has been reported (90%) in the general population [90], and findings by nature do not include individuals who have never conceived. Varying lengths of latent labour were reported by participants, with some reporting latent labour for much of their pregnancy, which might be better characterised as an irritable uterus. Some participants found it difficult to distinguish between the defined latent and active stages of labour; and so, responses were reliant on the participant’s recall. Further prospective research would be beneficial to objectively document the pregnancy and birth experiences of people with hEDS/HSD.

The interpretation of the data should also be considered in light of the sample demographics. The sample population was more highly educated than average, and aspects (e.g., cycle length) that may impact fertility were not measured [91]. The majority of participants were white; therefore, the findings of this study may not represent the experiences of people of other ethnicities in the included countries. This may have occurred because of the recruitment methods used, and future work should consider how to achieve a sample that is more reflective of the whole population. Availability and analysis of accurate record data (including improved diagnosis and recording of hEDS/HSD) is needed. Future statistical research may also consider the role of comorbid conditions and factors such as childbearing age or weight.

The word ‘woman’ was used during recruitment, which led to us being contacted by people who have experienced childbearing but do not identify as cisgender women and therefore felt excluded from the research. We did not collect gender demographics within this survey and so cannot say that all who took part were cisgender women, but this potential lack of representation for childbearing people as a whole group should be considered. Following this feedback, further research was carried out to examine perinatal care for trans and non-binary birthing people [92].

The majority of reported births occurred in obstetric units. This can typically be because of planned or emergency caesareans, or to manage birth complications or high-risk pregnancies. It is important to note, however, that people with hEDS/HSD should not automatically be considered to have a high-risk pregnancy, because of the wide heterogeneity in clinical presentation [28] with positive experiences and favourable birth outcomes also reported [23,93]. Many people with hEDS/HSD can still experience an uncomplicated pregnancy and birth and should not be discouraged from birthing their babies vaginally where appropriate [3]. As with every pregnant person, planning of perinatal care should be based on the individual’s situation, risks and wishes. Recommendations from this research focus on updating the awareness of possible complications and outcomes in healthcare guidance, and on including hEDS/HSD in initial perinatal screening questionnaires to ensure appropriate consultation and monitoring. The aim should now be to increase education, preparation, and available support for people with hEDS/HSD and perinatal staff, in order to reduce the risk of complications potentially causing an exacerbation into higher levels of disability. This is examined further in the second part of this programme of research [41] and information on the research programme can be found on this website for public dissemination of research: (www.hEDStogether.com).

## 5. Conclusions

These data cover a large number of pregnancies (*N* = 1338) over six countries and three continents, inclusive of up-to-date diagnostic criteria for hEDS/HSD. This research draws us closer to understanding how pregnancy and birth complications may be linked to hEDS/HSD. This is within the context of the recent understanding that 1 in 20 births could be affected, with many people with hEDS/HSD being undiagnosed. The findings from this study have highlighted that the following incidences were higher in people with hEDS/HSD than the general population: preterm birth, pre-eclampsia, eclampsia, PROM, antepartum haemorrhage, postpartum haemorrhage, hyperemesis gravidarum, shoulder dystocia, caesarean wound infection, postpartum psychosis, PTSD, precipitate labour and being born before arrival at intended place of birth. These can have profound impacts on perinatal and neonatal outcomes. Recommendations include updating healthcare guidance to include awareness of these possible complications and outcomes and including hEDS/HSD in initial screening questionnaires of perinatal care to ensure that appropriate consultation and monitoring can take place from the start. These findings also raise new questions about the aetiology of certain perinatal complications, which could lead us to a better understanding of causes and therefore treatment options for some serious and even life-threatening pregnancy-related conditions.

## Figures and Tables

**Figure 1 ijerph-20-06957-f001:**
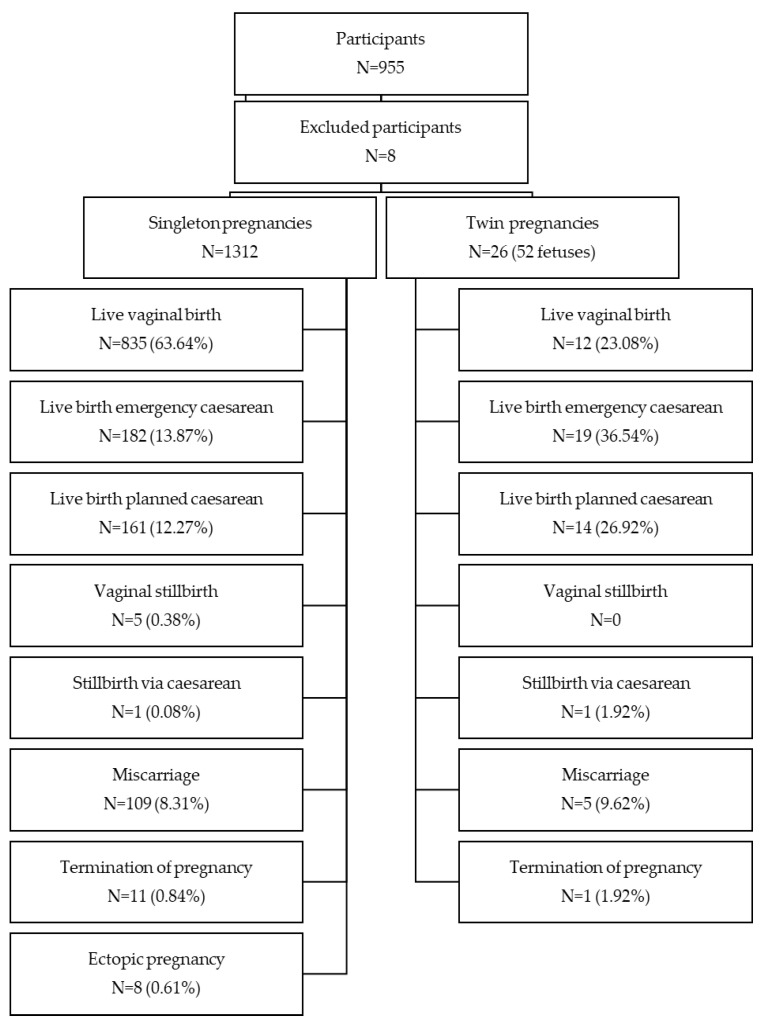
The total number (*N*) of participants, excluded participants and reported outcomes for singleton and twin pregnancies.

**Figure 2 ijerph-20-06957-f002:**
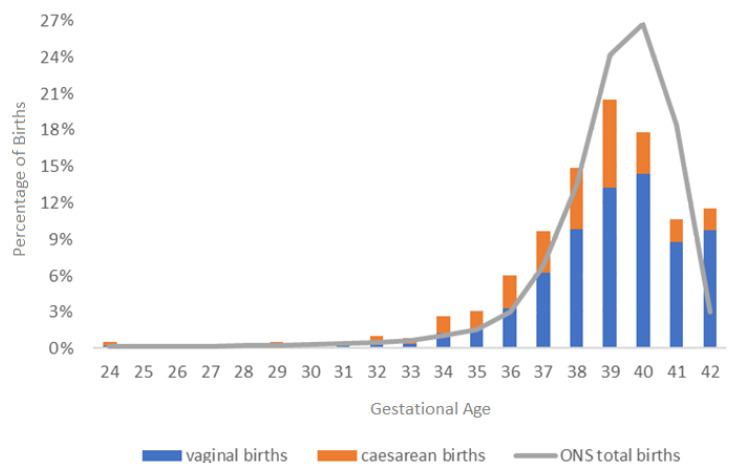
The percentages of vaginal and caesarean births for participants with hEDS/HSD (N = 1230), illustrated alongside the total births in England and Wales Office for National Statistics (ONS total births) [45] data by gestational age (week of birth).

## Data Availability

Raw data is not publicly available due to ethical restrictions. The corresponding author can be emailed with questions about the data. Publicly accessible information can be found about the programme of research at this website: (www.hEDStogether.com).

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
