# Peer review of "Childbearing with Hypermobile Ehlers–Danlos Syndrome and Hypermobility Spectrum Disorders: A Large International Survey of Outcomes and Complications"

_ijerph, 2023, doi:10.3390/ijerph20206957_

Round 1

Reviewer 1 Report

Please see the attached word file. 

Author Response

Response to Reviewer 1 (full response uploaded as attachment too)

This well-written article aims to give an overview of the outcomes and complications during pregnancy and after giving birth in women with hEDS or HSD. Data were are based on a large sample size and are compared with reference data and are well structured. Recommendations for clinical practice are described. As such, this research has a large clinical value and may benefit management of pregnant women with hEDS or HSD.

The authors express many thanks to the reviewer for taking the time to review our manuscript and for providing useful comments and suggestions to improve the manuscript. We agree that the research has significant clinical value, and that the large sample size received is indicative of people with hEDS/HSD wanting to strengthen research in this area. Thank you to the reviewer for their suggestions, which we have considered and made changes in response. Please find our changes and responses detailed below. Changes have also been highlighted in blue text in the re-uploaded manuscript.

 General comments: Although its strong clinical value, some general comments have to addressed:

1.1 REFERENCES: Please update the references, they don’t fit the content. They seem to be shifted.

We sincerely apologise for an error that occurred with the referencing software just prior to submission, which led to the order of references shifting so that in-text citations did not match to the correct numbered reference. The references have been corrected. These corrections resolve inconsistencies in how in-text claims are supported by references.

1.2 INCLUSION CRITERIA: Along the article, the authors use the diagnostic label ‘HSD’ though do not specify which type of HSD. To my opinion, most types of HSD (e.g., localized or peripheral HSD) are different from hEDS regarding symptoms and severity range and should therefore be excluded from this research

Evidence suggests that the diagnostic distinction between HSD and hEDS may not be clinically meaningful, with hEDS/HSD encompassing the same types of impairments and potential functional limitations. This is now further discussed, citing a recent 2022 extensive literature review [10]. Whilst individuals with hEDS/HSD may each experience different manifestations of the conditions, it was therefore not deemed appropriate based on recent evidence to reduce eligibility within HSD. In addition, diagnostic criteria prior to 2017 would not have distinguished between hEDS/HSD, and we incorporated previous names for these diagnoses in the recruitment eligibility criteria. The manuscript includes the following in the newly added box 1:

Individuals who do not meet the full hEDS criteria but who exhibit similar signs and symptoms may be diagnosed with a Hypermobility Spectrum Disorder [1,7]. Although a ”spectrum” of symptoms has been described, it is now understood that people with either hEDS or HSD may experience the same potential range of symptoms [8] complications and disease severity [9] and recommended treatments and prognosis are the same, thus the distinction may not be clinically meaningful [10-11].”

Additionally, in clinical practice, few patients seem to be told which HSD subtype they have (of those diagnosed since 2017). The 2017 hEDS criteria were specifically designed to be ”narrow” for the purposes of research into the genetic nature of the condition and importantly, were not validated prior to publication. The International Consortium is in the process of developing updated, validated clinical criteria, although this work may in part be superseded by genetic findings from the Norris Laboratory and the HEDGE study. The many iterations of the criteria for these conditions over time do make publishing research in this area complex, but our group is committed to continuing to do the best we can with imperfect historical labels.

1.3 STATISTICAL ANALYSES. the authors compared their data with reference data by descriptives. Please consider adding statistical analyses to further strengthen these findings.

Statistical examinations via inferential tests were not conducted for this study due to differences in the sources of data and data collections methods of this study (i.e., self-reported retrospective recall data) and population incidences (e.g., national medical record data including an unknown prevalence of hEDS/HSD diagnoses). These differences in data collection methods may contribute risk of bias to conducting statistical comparisons, and so to facilitate transparency and accuracy with the conclusions of this study, we instead present comparator figures to illustrate and highlight areas that would benefit from future research. The side-by-side presentation of descriptive statistics highlight the notably different prevalence rates of some complications. The discussion also includes the following:

“Higher incidences are concluded whereby ranges for population incidences fall entirely outside of the confidence intervals found in this study.” (line 269-271)

“Population health surveys are useful for estimating condition prevalence [88], and this research specifically provides novel insights into perinatal complications and outcomes that may need additional considerations by contextualising them with reported incidences in the general population. Comparison statistics were chosen with preference given to recently published large empirical studies, reviews or guidelines from a similar context (e.g., high income country) and international variation in outcomes such as gestational age have been reported [89]. However, comparator data were not available for all complications or nations, and these statistics cannot be interpreted as a control group. This examination does provide a clear need for future investigation, for example to collect and analyse large scale coded data, to document longitudinal childbearing journeys with hEDS/HSD, and to consider potential mechanisms for these associations.” (lines 371-382)

1.4 FIGURES/TABLES: please add legends explaining what is presented in the figures or tables and explaining abbreviations used.

 Further details have been added to Legends, and Figure 2 has been modified to improve the figure (removal of horizontal lines and amendment of axis labelling and titles).

Specific comments:

2.1 INTRODUCTION

Generally, the introduction needs an update of the references as they do not fit the content stated.

References have been corrected (1.1)

2.2 56-57: I would suggest to state this more carefully, as the diagnostic labels are based upon the structural systemic involvement and/or mendelian transmission within the spectrum of hypermobility. -Furthermore, there is large clinical heterogeneity.

Please see comment 1.2

The wording has been changed to: Although a ”spectrum” of symptoms has been described, it is now understood that people with either hEDS or HSD may experience the same potential range of symptoms [8] complications and disease severity [9] and recommended treatments and prognosis are the same, thus the distinction may not be clinically meaningful [10-11].” (box 1)

69-71: Add reference(s)

The sentence referred to by the reviewer was followed up with specific referenced examples. This paragraph has been revised to make this clearer and now reads:

“there are limitations to the existing evidence base, including that research has not always differentiated between hEDS and the other rare subtypes of EDS. To illustrate, a population-based study found higher rates of premature birth, birth by caesarean section and several other complications, along with longer hospital stays for post-partum care [32]. However, these findings incorporated all types of EDS, including the vascular subtype known to have serious risks during pregnancy [33].” (lines 55-60)

  • 78-80: As you mentioned this – it might be interesting to cover this topic in the manuscript

Gynaecological considerations are now considered throughout the discussion in light of the study’s findings, and existing research and practice guidelines.

MATERIALS AND METHODS

3.1 – 115-116: I’m not an expert, but by excluding those <24 weeks, you ‘miss’ miscarriages before 24 weeks?

We have clarified our writing to better communicate eligibility criteria that people had to have at least one birth after 24 weeks but could also report miscarriages before 24 weeks in separate question “rounds”:

“Participants recruited were women over the age of 18 years who had given birth after 24 weeks of pregnancy at least once from 2007 onwards (though participants meeting this criterion could also report on all pregnancies after 2007).” (lines 89-90)

“Participants repeated the questions for each pregnancy since 2007”. (line 116)

The limitation that people who had only experienced miscarriages would not have been eligible for the survey is included in the discussion:  

“Eligibility criteria and self-selection of volunteer participants should be considered in the context of the findings being based on retrospective self-report data. To reduce recall bias and sit within current guidance [30], the cut off year for pregnancies was 2007, however this means that maternity history provided may only be partial for some individuals. Although participants reporting a birth after 24 weeks could report miscarriages, individuals who have only experienced miscarriages were not eligible to take part.” (lines 383-389)

3.2 – 119: As mentioned above: which type of HSD? Consider only including those with the generalized type as this is most ‘similar’ to hEDS.

Please see comment 1.2. Additionally, of the 454 participants diagnosed under the 2017 criteria, only 17% had a diagnosis of HSD compared with 83% who met the full hEDS criteria. We did not ask which subtype of HSD participants had, so it would not be possible to remove those with peripheral or localised HSD. It is arguable that both generalized AND historical HSD are likely to be most similar to hEDS, but we will of course have to await further genetic and biomedical advances to be sure.

3.3 -135: consider statistical analyses

Please see comment 1.3

3.4 139: independently from each other?

Further detail about the qualitative analysis is now included between lines 131 and 136, including:

 “To increase reliability, two members of the research team were collectively involved in coding and analysis, with comments coded independently and then discussed with the second researcher (GP, LB). Qualitative codes, and associated quotes relating to complications of childbearing were grouped into themes. Themes were developed inductively from text, rather than from the complications directly asked about in the survey. Findings were discussed in light of perinatal care (SP) and medical expertise with lived experience (ER).”

RESULTS

4.1 - General comment: Are there any data on influencing factors (such as age and weight) of the mothers?

Weight was not collected and would be of interest for future studies, alongside age. The discussion now includes:

Availability and analysis of accurate record data (including improved diagnosis and recording of hEDS/HSD) is needed. Future statistical research may also consider the role of comorbid conditions, and factors such as childbearing age or weight.” (lines 404-407)

4.2 - 143-145: All eight? Strange that all eight made the same mistake?

The eight removed surveys were suspected to have entered four pregnancies as a single episode, indicated by the nature of open-ended responses from these participants. Our best guess is that these eight participants were attempting to give us data on pregnancies that occurred before 2007, but the online questionnaire did not permit this. Text has been updated for clarity to reads as follows:

Data for outcomes and complications are based on 947 responses because eight respondents appeared to enter data on four separate pregnancies in one single round of the survey and so were excluded from analysis.” (lines 144-146).

4.3 - 160-164: consider adding percentages

Percentages have been added to this section (lines 160-164) and also to the section above on education for consistency (lines 148-151).

4.4 - Figure 1: miscarriage: after 24 weeks?

        169: before 24 weeks were not recruited, why not N=1338?

We have clarified our writing in the methods to better communicate eligibility criteria that people had to have at least one birth after 24 weeks but could also report miscarriages before 24 weeks in separate question “rounds” (see answer to 3.1). Fetal demise after 24 weeks is reported within the stillbirth numbers.

4.5- 175-176: Are these data similar in the other included countries?

We used the ONS figures as they publish detailed annual data, and the largest proportion of participants (58%) were from the UK. A comment has been added to the discussion section:

Comparison statistics were chosen with preference given to recently published large empirical studies, reviews or guidelines from a similar context (e.g., high income country) and international variation in outcomes such as gestational age have been reported [89]. However, comparator data were not available for all complications or nations, and these statistics cannot be interpreted as a control group.” (lines 375-382)

4.6 - Figure 2: Add horizontal and vertical axis (gestational age and percentage of births)

Figure 2 has been amended to include axis labels.

4.7 - Table 1:

  • (31) (30): are these references? Why not between square brackets?

Citations changed to square brackets.

  • Given the more pronounced systemic involvement in hEDS compared to (G-)HSD: is there a difference in incidence of complications (e.g., pelvic organ prolapse) between those two groups?

Please see comment 1.2 and 3.2

4.8 - 221: typo – space between ‘described’ and ‘complications’

Changed to “Where participants selected ‘other’, listed complications included…” (line 227)

4.9 - 224-226: would be interesting to add incidences (%)?

We did consider this, however it was deemed that adding percentages would not be suitable because other participants were not prompted to think or report about those complications in the open-text questions, and therefore ascribing percentages to themes may inadvertently suggest prevalence rates within the sample that may not be accurate to reflect. The manuscript now includes, “Code frequencies are not reported with findings, as qualitative data were collected to supplement quantitative data and cannot infer prevalence of a complication.” (lines 139-141)

DISCUSSION

5.1- 274-277: Interesting to take a look at the data and see if indeed these mothers with bleeding also had pre-eclampsia? Consider statistical analyses?

This would be a very interesting analysis for future research, ideally with the use of accurate and complete medical records, but is out of scope for this study.

5.2 - 338: Do you have data about the prevalence of MCAS in this study cohort?

Data on MCAS or any other comorbid conditions were not collected. The questionnaire was quite long without adding questions on other conditions, and the survey was shortened following advice by ethical review. We agree that this would be a very interesting and important question.

- 342-343: Add example of treatment for MCAS

5.3 We have added an example of a treatment for MCAS: “…treatments for MCAS (e.g., identifying and removing environmental triggers, and the use of mast cell stabilisers and antihistamines) may be indicated alongside mainstream perinatal care [82].” (lines 340-342)

5.4- 353: Is there any research about the mechanisms of the higher incidence of hyperemesis gravidarum? Could autonomic dysfunction play a role?

Text added in relation to a suggested associated with POTS, with citation to literature, “Given that there is also an increased risk of hyperemesis gravidarum in women with PoTS, potential mechanisms might involve dysfunction of the autonomic nervous system [84].” (lines 351-353)

5.5- 426-427: Consider non-parametric testing to see if there were differences between countries

It was not our intent to look for differences between countries, but primarily to explore pregnancy outcomes in people with hEDS/HSD in developed nations. This sentence has therefore been removed from the discussion following a suggestion to make it more concise.

Reviewer 2 Report

Thank you for the opportunity to review this paper. This large survey (particularly in the context of the level of research in this area) reveals very interesting results. As the authors note, the over recruitment in a short period reveals that the participants feel the questions relate to an important area for them. As someone interested in supporting people with hypermobility it is important to see such results and it is also important to ensure the message from the paper is clear and well justified.

I suggest some major amendments, along with some minor ones. These major amendments are given with support in mind as I do think the authors have gathered useful information that should be published, however, at the moment the power of the results is unclear and the introduction is not yet well constructed.

I have some minor and more major comments that I hope will help to improve the paper.

Major

1.       I suggest a deep re working of the introduction. It covers a broad range of topics but is confused especially in relation to the use of some of the references. At the moment it doesn’t set out the problem well and struggles to get to the point.

The references are very confused as the authors sometimes refer to HSD/hEDS when the references don’t relate to this classification system. There is also confusion as the authors suggest that the severity of symptoms and types of symptoms are similar across HSD and hEDS – this is not the case. Indeed, the point of the hEDS criteria were to give some homogeneity to the top end of this symptomatic group so that a genetic marker might be more easily found. I wonder if there was some confusion and this statement actually relates to the similarity between EDS-HT/III and JHS - a very different classification to hEDS. Whatever the reason, I suggest that the confusions related to statements about HSD/hEDS which have been referenced to data using a different classification system; this needs to be ironed out.

I am also not convinced by the references relating to prevalence of HSD and hEDS combined, where the authors suggest that there is a high prevalence in the general population. L38 reference 5 doesn’t seem to link to the data described in the statement. References 6 and 7 don’t relate to hEDS/HSD and I can’t therefore see a link to the statement supported by reference 8. Certainly asymptomatic hypermobility is prevalent but symptomatic HSD and especially hEDS in a general population is not high. I do not think this is the argument to make when justifying its importance. When tracking back through the references to try and find the original data that suggests this prevalence I got lost. I think that data referenced within these papers isn’t the original data. When trying to find the original data I think they included asymptomatic hypermobility. Therefore, if this is to be an argument for the importance of the work, then this should be referenced from stronger and original data which is designed to explore prevalence. This doesn’t mean that this isn’t an important population. I’m just not convinced by the references presented in the introduction and discussion.

There is also confusion further into the paper when it speaks of different subtypes of EDS eg in L69 and in the discussion L268. A naive reader might think that the HSD and hEDS are the subtypes you mean but that is not what the authors are referring to. I suggest that laying out that EDS has different sub-types in more detail earlier in the Introduction would make reference to this clearer.

Your 3rd and 4th paragraphs include a deeper analysis of each paper than required in an introduction. Please synthesis the key messages to justify the need for this work.

I would suggest a reworked introduction starts with: what EDS is with the different sub-types. Then, that one subtype doesn’t have a genetic marker and is on a spectrum. The spectrum is wide and at the top end is hEDS with specific criteria of xyz. The other parts of the spectrum only require symptomatic hypermobility. For HSD, some of the symptoms might include one or more of xyz. Previous work has suggested that people with symptomatic hypermobility (a phrase to get around the different classification systems used in this work) suffer with complications when giving birth. To date these complications include xyz however, the work is not current and can be improved because of xyz. Therefore we undertook this work in order to …… So simplify the story to make it clearer to read. I hope that this helps.

2.       Methods: More description of the methods is needed. How was the survey constructed – was there a framework used or a process ie a literature search to understand what is currently known, a PPI group to develop the concepts to explore, a stake holder group to develop questions from these concepts, piloting etc, timing of the process and how long the survey was open. This will then explain how you got to the final survey.

3.       The methods of analysis was very under described and didn’t include what you described in the results. Please go through your results and ensure the plan includes all that you’ve done in the methods.

4.       You suggested you had a power calculation as you write that the survey went over the number needed, but this isn’t described in the methods

5.       Results: I’m very confused by the analysis. This might be because I wasn’t sure of the outcomes used to measure some of the content. An appendix or supplementary material including the survey so the outcomes can be understood would help. In particular, but not alone, I can’t work out how 95% CI were constructed from some of the data, some of which isn’t likely to be normally distributed. Greater detail in the analysis plan might help me understand this.

6.       The results suggest that the data was ‘compared’ and was ‘significantly’ different to some population data. How was significance defined? It is not always clear what normative data was used and therefore the reader can’t judge whether they trust that ‘comparison’. I think they haven’t compared the data statistically, therefore I would be careful about how this is expressed. For example one might say that - for illustrative purposes it is interesting to note the data from x source seems to differ. If instead you would like to suggest statistical difference by saying the confidence intervals are outside the specific population data, far more detail is needed in both the methods to explain that and the illustration. The illustration is poor - there are no labels for axes, no reason to have horizontal lines going through the data, % at 2 decimal points when this detail is not going to be seen on a graph such as this

7.       In the discussion you ask what complications are related to hEDS/HSD. I’d avoid the word related as your methods didn’t look for relationships statistically

8.       The discussion is overly long and I suggest needs to be reduced so that the main messages can shine through.

9.       One surprising omission which I think is very important is a discussion on the population in relation to ethnicity. Your population was highly white. This is very biased as, in particular, the population of people with HSD/hEDS is impacted by ethnicity. Therefore one would expect that if the surveyed population reflects the population of the people with this condition, it would be of a mixed ethnic origin. A discussion related to the educated, white participants needs to be put in context with the bias that could be embedded through the methods of recruitment. If we are to understand the issues that this cohort face, we need to ensure we recruit people that reflect this cohort of individuals.

Minor

L57 “In the current 2017 criteria [2], all previous diagnosis classifications of EDS Type III [18], EDS - Hypermobility Type [19] or Joint Hypermobility Syndrome (JHS) [20] were deemed to be encompassed by hEDS/HSD without the need for reassessment, hence this current study refers to hEDS/HSD throughout inclusive of these previous diagnoses.” There is no reference to this statement.

L61 “The multisystemic nature of connective tissue means that hEDS/HSD most commonly manifest as pain and fatigue, and have gynaecological, urological, gastroenterological, neurological, cardiovascular, autonomic and immunological manifestations [21].” This might leave the reader thinking that people with HSD suffer with all of these things. Please re phrase as they definitely could but more often don’t.

L69 “Existing research on the complications of pregnancy in EDS commonly fails to differentiate between different subtypes of EDS, has small numbers of participants or recruits  through specialist clinics with a bias towards more severely affected individuals” Please reference. In addition, the different subtypes have not been clearly listed in the introduction to make clear that you aren’t referring to the different HSD sub-types and hEDS.

L201 qualitative statements. Statements will be qualitative so no need to say ‘qualitative’.

L205 adds to the confusion related to your statistical analysis. You have said the data it isn’t normally distributed (why? How was this discovered?) but then talk about standard deviations (SD). How are SDs relevant if the data is normally distributed?

L221 minor typo ‘described-complications’

L225 an example of a lack of information related to the analysis. Five themes were given but I’m unclear how the themes were derived as nothing about this in the methods.

L267 sentence construction could be revised to improve clarity and flow

L306 ‘it may also be useful to consider’ – Why? This may lay out the importance of the work.

L359 unclear why your data corroborates this topic

L371 I’m not sure that illness best describes this condition. I’d use the word condition

The clarity of the paper is not related to the use of English, but more the construction and structure. Please see above

Author Response

Response to Reviewer 2 (attached full response to reviewers too)

Thank you for the opportunity to review this paper. This large survey (particularly in the context of the level of research in this area) reveals very interesting results. As the authors note, the over recruitment in a short period reveals that the participants feel the questions relate to an important area for them. As someone interested in supporting people with hypermobility it is important to see such results and it is also important to ensure the message from the paper is clear and well justified.

I suggest some major amendments, along with some minor ones. These major amendments are given with support in mind as I do think the authors have gathered useful information that should be published, however, at the moment the power of the results is unclear and the introduction is not yet well constructed.

I have some minor and more major comments that I hope will help to improve the paper.

The authors express many thanks to the reviewer for taking the time to review our manuscript and for providing useful comments and suggestions to improve the manuscript. Please find our changes and responses detailed below. Changes have also been highlighted in the re-uploaded manuscript.

  • I suggest a deep re-working of the introduction. It covers a broad range of topics but is confused especially in relation to the use of some of the references. At the moment it doesn’t set out the problem well and struggles to get to the point.

The introduction has been reordered in line with the reviewer’s suggestions for a format. We hope that this restructuring better sets out the story, leading up to the value of the research. We have also created a section in box 1 relating to the history of nomenclature and diagnostic criteria for hEDS/HSD.

The references are very confused as the authors sometimes refer to HSD/hEDS when the references don’t relate to this classification system. There is also confusion as the authors suggest that the severity of symptoms and types of symptoms are similar across HSD and hEDS – this is not the case. Indeed, the point of the hEDS criteria were to give some homogeneity to the top end of this symptomatic group so that a genetic marker might be more easily found. I wonder if there was some confusion and this statement actually relates to the similarity between EDS-HT/III and JHS - a very different classification to hEDS. Whatever the reason, I suggest that the confusions related to statements about HSD/hEDS which have been referenced to data using a different classification system; this needs to be ironed out. I am also not convinced by the references relating to prevalence of HSD and hEDS combined, where the authors suggest that there is a high prevalence in the general population. L38 reference 5 doesn’t seem to link to the data described in the statement. References 6 and 7 don’t relate to hEDS/HSD and I can’t therefore see a link to the statement supported by reference 8. Certainly asymptomatic hypermobility is prevalent but symptomatic HSD and especially hEDS in a general population is not high. I do not think this is the argument to make when justifying its importance. When tracking back through the references to try and find the original data that suggests this prevalence I got lost. I think that data referenced within these papers isn’t the original data. When trying to find the original data I think they included asymptomatic hypermobility. Therefore, if this is to be an argument for the importance of the work, then this should be referenced from stronger and original data which is designed to explore prevalence. This doesn’t mean that this isn’t an important population. I’m just not convinced by the references presented in the introduction and discussion.

We sincerely apologise for an error that occurred with the referencing software just prior to submission, which led to the order of references shifting so that in-text citations did not match to the correct numbered reference. The references have been corrected. These corrections resolve inconsistencies in how in-text claims are supported by references. We have also added references to the fact that HSD can be just as impactful as hEDS in box 1.

“Although a ”spectrum” of symptoms has been described, it is now understood that people with either hEDS or HSD may experience the same potential range of symptoms [8] complications and disease severity [9] and recommended treatments and prognosis are the same, thus the distinction may not be clinically meaningful [10-11].”

There is also confusion further into the paper when it speaks of different subtypes of EDS eg in L69 and in the discussion L268. A naive reader might think that the HSD and hEDS are the subtypes you mean but that is not what the authors are referring to. I suggest that laying out that EDS has different sub-types in more detail earlier in the Introduction would make reference to this clearer.

Thank you for your helpful comments. The introduction has been restructured according to your suggestions and the section in the introduction that was on line 69 has been changed as follows:

However, there are limitations to the existing evidence base, including that research has not always differentiated between hEDS and the other rare subtypes of EDS. To illustrate, a population-based study found higher rates of premature birth, birth by caesarean section and several other complications, along with longer hospital stays for post-partum care [32]. However, these findings incorporated all types of EDS, including the vascular subtype known to have serious risks during pregnancy [33].” (lines 55-60)

The discussion has been revised and explicit reference to “EDS as a whole” has been removed based on reviewer suggestions for conciseness.

Your 3rd and 4th paragraphs include a deeper analysis of each paper than required in an introduction. Please synthesis the key messages to justify the need for this work.

These two paragraphs have been synthesised into one as follows:

Existing research is inconsistent about the rates of obstetric complications among people with hEDS/EDS. Increased rates of miscarriages (including specifically recurrent miscarriage) [34], preterm birth [35], and pelvic instability [36], have been reported, whilst a large records-based study found pregnancy complication rates similar to population incidences [37]. A small but in-depth study of women with JHS found “good” overall outcomes, but potential issues with abnormal scar formation, haemorrhage, pelvic prolapses, deep venous thrombosis, and coccyx dislocation [38]. Precipitate labour (the expulsion of the fetus within three hours of commencement of contractions) [39] has also been suggested as possibly more prevalent among people with hEDS/HSD [23,38], and is associated with significant complications [40]. Qualitative research has reported experiences of worsening of hEDS/HSD symptoms during pregnancy, as well as ineffective local anaesthesia and long latent phases of labour followed by rapid births [23]. Contemporary evidence with larger sample sizes following the 2017 re-classification [1] is now required to enhance the quality and safety of evidence-based practice. The aim of this study was to examine pregnancy and birth outcomes in people with hEDS/HSD through a large-scale international survey; to provide a new understanding of the full range of complications and thereby inform perinatal care and increase obstetric safety.” (lines 61-77)

I would suggest a reworked introduction starts with: what EDS is with the different sub-types. Then, that one subtype doesn’t have a genetic marker and is on a spectrum. The spectrum is wide and at the top end is hEDS with specific criteria of xyz. The other parts of the spectrum only require symptomatic hypermobility. For HSD, some of the symptoms might include one or more of xyz. Previous work has suggested that people with symptomatic hypermobility (a phrase to get around the different classification systems used in this work) suffer with complications when giving birth. To date these complications include xyz however, the work is not current and can be improved because of xyz. Therefore we undertook this work in order to …… So simplify the story to make it clearer to read. I hope that this helps.

The introduction has been restructured based on your helpful suggestion (lines 31-77).

  1. Methods: More description of the methods is needed. How was the survey constructed – was there a framework used or a process ie a literature search to understand what is currently known, a PPI group to develop the concepts to explore, a stake holder group to develop questions from these concepts, piloting etc, timing of the process and how long the survey was open. This will then explain how you got to the final survey.

Further details have been added to the Method section with respect to survey construction and date collection as follows:

“Participants were recruited through public, professional, voluntary sector and social media platforms between June and September 2019.

The survey questions were developed following review of the available literature in relation to childbearing with hEDS/HSD, and modified following piloting and ethical review (e.g., questions were prioritised to reduce survey length). Patient and public involvement (PPI) was conducted via an online poll and led to the inclusion of certain complications, for example, premature birth, abnormal length of labour/birth, poor anaesthetic coverage and haemorrhage (see Pearce et al., 2023 for further detail of PPI [41]). To facilitate understanding of terms (e.g., augmentation and induction of labour), definitions were provided to participants and are reported here alongside the results. The survey was hosted on Qualtrics software starting with participant information and consent before completion of the survey. Complex survey logic was implemented so participants were not asked to complete inappropriate or unethical questions (e.g., after reporting a miscarriage).” Lines 97-110

 “Participants reported their demographics (age, ethnicity, level of education), and for each pregnancy, reported the birth outcome, gestational week, lengths of birth stages, place and mode of birth, and then selected which birth complications they had experienced from a list (see table 1). Participants repeated the questions for each pregnancy since 2007, and could provide any further comments about their childbearing experience via an open text box.” (lines 113-118)

Acknowledgement has been added to PPI:

“We would like to thank all those who completed the patient and public involvement online poll and informed the development of the survey.” (lines 467-468)

  1. The methods of analysis was very under described and didn’t include what you described in the results. Please go through your results and ensure the plan includes all that you’ve done in the methods.

The method of analysis now provides more detail as follows:

Data were analysed with SPSS 28 software using descriptive statistics to two decimal places. Data were reviewed to ensure all responses met eligibility criteria and descriptive statistics were calculated for participant demographics. Percentage incidences were calculated for outcomes and complications (e.g., pre-eclampsia, pre-term birth, shoulder dystocia, haemorrhages, ineffective pain relief). Where responses were binary, i.e., experienced or not experienced, 95% confidence intervals for proportions were calculated. Survey data are presented alongside relevant published general population incidences, collated where available following literature review, to aid interpretation in the context of the study design.

All qualitative responses were read for familiarity and analysed using conventional content analysis to compliment the quantitative focus and capture additional complications not asked about quantitatively. This approach is appropriate when limited literature exists and to analyse alongside descriptive statistics [43-44]. To increase reliability, two members of the research team were collectively involved in coding and analysis, with comments coded independently and then discussed with the second researcher (GP, LB). Qualitative codes, and associated quotes relating to complications of childbearing were grouped into themes. Themes were developed inductively from text, rather than from the complications directly asked about in the survey. Findings were discussed in light of perinatal care (SP) and medical expertise with lived experience (ER). Code frequencies are not reported with findings, as qualitative data were collected to supplement quantitative data and cannot infer prevalence of a complication. Any identifiable information in open-ended responses was anonymised.” (Lines 121-142)

  1. You suggested you had a power calculation as you write that the survey went over the number needed, but this isn’t described in the methods

This line has been reworded for clarity, “The number of completed surveys received was 955.” (line 144). This paper is the first part of a larger project that explored childbearing women and healthcare professionals’ experiences of perinatal care (see line 84-87), and a power analysis was conducted to enable statistical analysis related to this second part of the project [41].

  1. Results: I’m very confused by the analysis. This might be because I wasn’t sure of the outcomes used to measure some of the content. An appendix or supplementary material including the survey so the outcomes can be understood would help. In particular, but not alone, I can’t work out how 95% CI were constructed from some of the data, some of which isn’t likely to be normally distributed. Greater detail in the analysis plan might help me understand this.

This has been addressed by adding more detail in the methods section. See answer to number 3 above.

  1. The results suggest that the data was ‘compared’ and was ‘significantly’ different to some population data. How was significance defined? It is not always clear what normative data was used and therefore the reader can’t judge whether they trust that ‘comparison’. I think they haven’t compared the data statistically, therefore I would be careful about how this is expressed. For example one might say that - for illustrative purposes it is interesting to note the data from x source seems to differ. If instead you would like to suggest statistical difference by saying the confidence intervals are outside the specific population data, far more detail is needed in both the methods to explain that and the illustration.

Language has been amended throughout the manuscript (including in the abstract, results, and discussion) to clarify that survey data were not statistically compared to population incidences, such as removal of the word “significantly” and instead using words like “higher” when referencing the typical population figures. We have also stated:

“Note. Outcomes and complications that have been put in bold highlight where confidence interval findings from this study sit outside of the general population incidences.” (table 1).

 “Higher incidences are concluded whereby ranges for population incidences fall entirely outside of the confidence intervals found in this study.” (lines 269-271)

Calls for updates to practice and future research are also made, “This examination does provide a clear need for future investigation, for example to collect and analyse large scale coded data, to document longitudinal childbearing journeys with hEDS/HSD, and to consider potential mechanisms for these associations.” (lines 379-382)

The illustration is poor - there are no labels for axes, no reason to have horizontal lines going through the data, % at 2 decimal points when this detail is not going to be seen on a graph such as this

Figure 2 has been modified by removal of horizontal lines and amendment of axis labelling and decimal places.

  1. In the discussion you ask what complications are related to hEDS/HSD. I’d avoid the word related as your methods didn’t look for relationships statistically

The word “related” has been changed to “linked” (line 436)

  1. The discussion is overly long and I suggest needs to be reduced so that the main messages can shine through.

The discussion has been reduced and reworked to better demonstrate the key messages. Though additional discussion points (e.g., limitations) are included to address reviewer feedback, overall word count for the discussion has been reduced.

  1. One surprising omission which I think is very important is a discussion on the population in relation to ethnicity. Your population was highly white. This is very biased as, in particular, the population of people with HSD/hEDS is impacted by ethnicity. Therefore one would expect that if the surveyed population reflects the population of the people with this condition, it would be of a mixed ethnic origin. A discussion related to the educated, white participants needs to be put in context with the bias that could be embedded through the methods of recruitment. If we are to understand the issues that this cohort face, we need to ensure we recruit people that reflect this cohort of individuals.

 Thank you for this suggestion, we have included this in the discussion:

 “Interpretation of the data should also be considered in light of the sample demographics. The sample population was more highly educated than average, and aspects (e.g., cycle length) that may impact fertility were not measured [91]. Participants were majority white, and therefore the findings of this study may not represent the experiences of people of other ethnicities in the included countries. This may have occurred because of the recruitment methods used and future work should consider how to achieve a sample that is more reflective of the whole population. Availability and analysis of accurate record data (including improved diagnosis and recording of hEDS/HSD) is needed.  ” (lines 398-406)

Minor

L57 “In the current 2017 criteria [2], all previous diagnosis classifications of EDS Type III [18], EDS - Hypermobility Type [19] or Joint Hypermobility Syndrome (JHS) [20] were deemed to be encompassed by hEDS/HSD without the need for reassessment, hence this current study refers to hEDS/HSD throughout inclusive of these previous diagnoses.” There is no reference to this statement.

References have been corrected following error. This section of the introduction has been restructured to provide further context, supported by references.

L61 “The multisystemic nature of connective tissue means that hEDS/HSD most commonly manifest as pain and fatigue, and have gynaecological, urological, gastroenterological, neurological, cardiovascular, autonomic and immunological manifestations [21].” This might leave the reader thinking that people with HSD suffer with all of these things. Please re phrase as they definitely could but more often don’t.

The introduction has been restructured and the above sentence is no longer included as quoted.

L69 “Existing research on the complications of pregnancy in EDS commonly fails to differentiate between different subtypes of EDS, has small numbers of participants or recruits through specialist clinics with a bias towards more severely affected individuals” Please reference. In addition, the different subtypes have not been clearly listed in the introduction to make clear that you aren’t referring to the different HSD sub-types and hEDS.

The introduction has been restructured to provide further clarity about the historical changes in diagnosis of hEDS/HSD, previous terms, current limitations of the evidence base, and the rationale for this research supported by references.

L201 qualitative statements. Statements will be qualitative so no need to say ‘qualitative’.

Change made to remove “qualitative”.

L205 adds to the confusion related to your statistical analysis. You have said the data it isn’t normally distributed (why? How was this discovered?) but then talk about standard deviations (SD). How are SDs relevant if the data is normally distributed?

Non-normality of data related to the latent phase of labour was discovered via visual observation on a histogram and non-normal skewness and kurtosis values. The manuscript now reads, “The data on latent labour were not normally distributed, as indicated on a histogram, with reports of occurrences that lasted weeks, and 148 (12.24%) reporting latent labours longer than 24 hours.” (lines 203-205)

L221 minor typo ‘described-complications’

Changed to “Where participants selected ‘other’, listed complications included…” (line 227)

L225 an example of a lack of information related to the analysis. Five themes were given but I’m unclear how the themes were derived as nothing about this in the methods.

Further detail has been added about the analysis. “All qualitative responses were read for familiarity and analysed using conventional content analysis to compliment the quantitative focus and capture additional complications not asked about quantitatively. This approach is appropriate when limited literature exists and to analyse alongside descriptive statistics [43-44]. To increase reliability, two members of the research team were collectively involved in coding and analysis, with comments coded independently and then discussed with the second researcher (GP, LB). Qualitative codes, and associated quotes relating to complications of childbearing were grouped into themes. Themes were developed inductively from text, rather than from the complications directly asked about in the survey. Findings were discussed in light of perinatal care (SP) and medical expertise with lived experience (ER). Code frequencies are not reported with findings, as qualitative data were collected to supplement quantitative data and cannot infer prevalence of a complication. Any identifiable information in open-ended responses was anonymised.” (lines 130-142)

L267 sentence construction could be revised to improve clarity and flow

Changes made, text now reads, “Whilst previous research has found a higher rate of miscarriage among people with hEDS [34] than has been reported in the general population [60], this study found no increased incidence.” (lines 279-281)

L306 ‘it may also be useful to consider’ – Why? This may lay out the importance of the work.

Restructured to read: “Prolonged latent phases of labour may quickly progress into fast active labours and births for people with hEDS/HSD, with subsequent risk that births are unattended by professionals. This outcome may contribute to a more negative birthing experience [69], as well as contribute to explaining the higher prevalence of pelvic organ prolapse (vaginal wall or womb) [70-71].”(lines 308-312)

L359 unclear why your data corroborates this topic

Text now reads, “Post-traumatic stress disorder (PTSD) was reported by almost 1 in 5 people in this study, consistent with the prevalence of PTSD previously found among high-risk groups (e.g., women who experienced a difficult or traumatic birth or had emergency caesarean sections) [59].”(lines 360-365)

L371 I’m not sure that illness best describes this condition. I’d use the word condition

Reworded to “condition”. (line 367)

Round 2

Reviewer 2 Report

I am very impressed by the quick turn around and improvements to the paper. It is much clearer and now includes detail to help the reader understand the strength of the evidence presented.